# Visualization of supercritical water *pseudo*-boiling at Widom line crossover

Florentina Maxim [1,2], Cristian Contescu[3], Pierre Boillat[4,5], Bojan Niceno[6,7], Konstantinos Karalis [6], Andrea Testino [1] & Christian Ludwig [1,8]

Supercritical water is a green solvent used in many technological applications including materials synthesis, nuclear engineering, bioenergy, or waste treatment and it occurs in nature. Despite its relevance in natural systems and technical applications, the supercritical state of water is still not well understood. Recent theories predict that liquid-like (LL) and gas-like (GL) supercritical water are metastable phases, and that the so-called Widom line zone is marking the crossover between LL and GL behavior of water. With neutron imaging techniques, we succeed to monitor density fluctuations of supercritical water while the system evolves rapidly from LL to GL as the Widom line is crossed during isobaric heating. Our observations show that the Widom line of water can be identified experimentally and they are in agreement with the current theory of supercritical fluid *pseudo*-boiling. This fundamental understanding allows optimizing and developing new technologies using supercritical water as a solvent.

[1] Laboratory for Bioenergy and Catalysis (LBK), ENE Division, Paul Scherrer Institute, 5232 Villigen PSI, Switzerland. [2] Laboratory for Chemical Thermodynamics, "Ilie Murgulescu" Institute of Physical Chemistry, Splaiul Independentei 202, 060021 Bucharest, Romania. [3] Materials Science and Technology Division, Oak Ridge National Laboratory, One Bethel Valley Road, Oak Ridge, TN 37831-6087, USA. [4] Electrochemistry Laboratory (LEC), ENE Division, Paul Scherrer Institute, 5232 Villigen PSI, Switzerland. [5] Laboratory for Neutron Scattering and Imaging (LNS), NUM Division, Paul Scherrer Institute, 5232 Villigen PSI, Switzerland. [6] Laboratory for Scientific Computing and Modelling (LSM), NES Division, Paul Scherrer Institute, 5232 Villigen PSI, Switzerland. [7] Eidgenössische Technische Hochschule Zürich (ETHZ), MAVT-LKE, 8092 Zurich, Switzerland. [8] École Polytechnique Fédérale de Lausanne (EPFL), ENAC IIE GR-LUD, 1015 Lausanne, Switzerland. Correspondence and requests for materials should be addressed to F.M. (email: fmaxim@icf.ro) or to C.L. (email: christian.ludwig@psi.ch)

Water in its supercritical state has unique properties, reflecting both the liquid and gas behavior, which makes it useful for applications, such as wastewater treatment[1], thermochemical conversion of biomass for biofuel production[2–5], and hydrothermal synthesis of nanoparticles[6,7]. In addition, supercritical water occurs in nature and it is currently an attractive geothermal resource for power production[8]. However, due to the cost and difficulty of undergoing processes at supercritical conditions, supercritical water technologies implemented on large scale are still far. The cost is high because the net energy balance is not always favorable[2] and the challenge to separate valuable products from fast reactions under supercritical conditions[9]. Optimizing the operation parameters for energy savings and efficient separation is of great importance for the development of supercritical water technologies. There is still missing fundamental knowledge on how the water physical properties change in supercritical states, knowledge needed to establish the optimal operating conditions.

According to textbooks[10,11], supercritical water is a single, homogenous phase present above the critical temperature ($T_{CP} = 647$ K) and the critical pressure ($P_{CP} = 221$ bar). In the van der Waals theory of criticality[12], the critical point is the end of the liquid–gas equilibrium (coexistence) curve, the point in which liquid and gas merge into each other in a continuous manner and reach the same density, the critical density ($\rho_{CP} = 322$ kg m$^{-3}$)[13].

Near the critical point, at temperatures from 600 to 700 K and pressures from 220 to 300 bar, the physical properties of water present drastic changes. For instance, the density decreases from the values for compressible liquid (about 750 kg m$^{-3}$) to values for dense gas (about 150 kg m$^{-3}$)[13]. These density variations near the critical point correlate with other macroscopic properties (molecular diffusivity, viscosity, and dielectric constant) and reflect changes at molecular level, such as molecular association by hydrogen bonding[2,14]. Carrying out processes in compressible liquid water or dense water vapor may result in energy savings and efficient separations[2,15].

There was a vivid debate on whether there is a liquid–gas phase change associated to the fluid density variations in the region near the critical point, as no macroscopically visible interface between liquid and gas appears in this region. In 1965 Widom demonstrated that, below and near the critical point, the liquid/gas interface thickness is equal to the molecular correlation length of fluid density fluctuations[16,17]. Moreover, in the supercritical region, some of the fluid thermophysical properties (isobaric heat capacity, thermal expansion coefficient, isothermal compressibility) show maxima, with the highest values close to the critical point[13]. Due to the fact that these maxima cannot be associated to a first-order liquid–gas phase transition (see Chapter 4.7 in ref. [11]), they are called critical anomalies[14]. In 1972, Widom demonstrated that the representation of the critical anomalies in the pressure–temperature space are lines emanating from the critical point[18]. Below a reduced pressure ($P_r = P/P_{CP}$) of 1.5 these lines of maxima in thermodynamic response functions converge on a single line as the liquid–gas critical point is approached[19]. When isobaric processes are analyzed, the most referred Widom line is the one indicating the locus of maxima isobaric heat capacity[20], as shown in Fig. 1a.

It was only in 2010 when Simeoni et al. [21] demonstrated by inelastic X-ray scattering and molecular dynamics in supercritical argon that the Widom line divides the supercritical region into two domains that, although not connected by a first-order singularity, can be identified by different dynamical regimes: gas-like and liquid-like. The same year, 2010, McMillan and Stanley associated this crossover at the Widom line with nanoscale density fluctuations of the supercritical fluid[22]. In 2014, Gallo, Corradini and Rovere showed, by using molecular dynamics simulations validated against NIST reference data[13], that also the supercritical state of water can be divided into a gas-like region and a liquid-like region by the Widom line anticipating the phase separation and the coexistence that is found below the critical point[23].

In 2018, Ha et al. proved by machine-learning analysis on density fluctuations simulation data that the GL and LL supercritical fluid co-exist also above the critical point, although the LL–GL equilibrium condition is not fulfilled[24]. Moreover, the same authors showed that LL and GL molecular units undergo continuous transition across a region that is enclosing the Widom line. The shape of this region resembles a deltoid that originates from the critical point and therefore should be called Widom delta[24].

Based mainly on the above-mentioned studies, now everyone accepts that LL and GL supercritical fluid are two phases distinguishable at molecular scale. Only recently, it became clear that the Widom line in the supercritical region obeys similar law like the coexistence lines below the critical point[25]. Moreover, the transition from LL to GL supercritical states when crossing the Widom line finds explanation based on the *pseudo*-boiling concept[26].

Up to date, there is no experimental evidence showing that LL and GL supercritical water are distinguishable at millimeters scale. The deficiency of experimental evidences is mainly due to the lack of reliable experimental techniques for studying in situ the water density fluctuations under supercritical conditions. Due

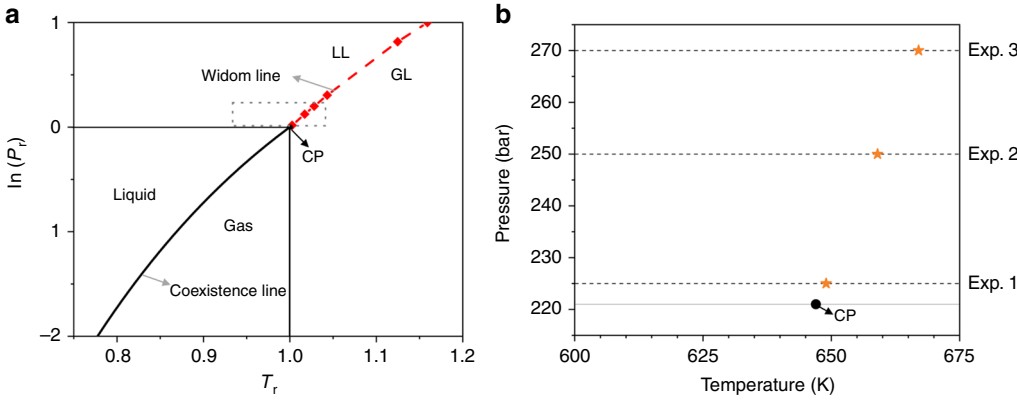

**Fig. 1** Water phase diagram indicating the supercritical region analyzed in this article. **a** Water states in the plan of reduced pressure ($P_r = P/P_{CP}$) and reduced temperature ($T_r = T/T_{CP}$); coexistence line below critical point (CP) separates liquid and gas phases and Widom line above CP separates LL and GL supercritical water states; **b** the region of our experiments in pressure–temperature space; the orange symbols indicate the points of the *pseudo*-boiling line[25] for each isobar. Source data are provided as a Source Data file

to the high neutron scattering cross section of a hydrogen atom[27,28], neutron imaging is a sensitive experimental technique for water density analysis[29].

Here, we analyze the density fluctuations of water flow passing through a porous medium made from activated carbon fibers[30]. Using neutron imaging, we visualize the dynamic conversion from LL to GL supercritical water and the evolution of the Widom line while increasing the temperature along three isobars near the critical point, as indicated in Fig. 1b. We discuss the results taking into account the water–solid carbon interaction during the supercritical water *pseudo*-boiling.

## Results

### Density fluctuations in the presence of carbon porous material.
A monolithic structure made of carbon fibers composite molecular sieves[30] was used as porous medium (referred hereafter as monolith). The monolith is a light body for its volume, with large volume of interfibrillar space (voids) and high permeability. Monolith's fibers have internal micropores and the large internal surface exposed in carbon micropores is hydrophobic, a common feature for this type of materials[31,32]. Our gas adsorption and scanning electron microscopy (SEM) analyses confirmed that there are two pore sizes to take into consideration in the monolith: micropores with size < 2 nm, and interfibrillar voids in the micrometers scale. The "Methods" section and Supplementary Information provide more details about the carbon fibers monolith.

We used neutron-imaging techniques to measure the water density fluctuations inside a tubular reactor containing the monolith, which was continuously passed through by an upwards water flow with a constant rate (Fig. 2a). In the field of view, we obtained color-scaled density images (Fig. 2b) and analyzed vertical density profiles (Fig. 2c). The "Methods" section presents the experimental details and the image-processing procedure.

The color-scaled density image in Fig. 2b shows the contrast between LL and GL. Supercritical water with LL densities ($\rho_{WL} < \rho_{LL} < 650\ \mathrm{kg\,m^{-3}}$) appears reddish and supercritical water with GL densities ($100\ \mathrm{kg\,m^{-3}} < \rho_{GL} < \rho_{WL}$) is yellowish (where $\rho_{WL}$ is the

density at the Widom line). At the end of the density scale, pressurized liquid water with densities up to $750\ \mathrm{kg\,m^{-3}}$ is blueish and the water vapor with a density approaching $0\ \mathrm{kg\,m^{-3}}$ is white. Figure 2b is an example of a density map recorded at some moment in the experiment, which represents an average of density in the direction of the beam, the plan normal to the fluid flow. It is evident that red and yellow regions co-exist in the zone of the monolith in dynamic evolution, although without a clear demarcation line between these two regions. Figure 2c is the water density contour obtained from Fig. 2b. It can be observed that $\rho_{before} < \rho_{monolith} < \rho_{after}$, where $\rho_{before}$, $\rho_{monolith}$, and $\rho_{after}$ are the density values before, through, and after the monolith. Moreover, while the water density value is constant before and after the monolith, through the monolith there is not a constant density value. The $\rho_{monolith}$ density profile presents a wave in the volume of the monolith, where in Fig. 2b the transition from yellow to red region is visible, meaning the transition from GL densities to LL densities.

The image in Fig. 2d presents the tomographic reconstruction within the region of the monolith of the density map presented in Fig. 2b. Based on this we computed the density histogram (Fig. 2e) showing the volume fraction of min and max densities values in the region of the monolith, where maximum is referring to LL densities and the minimum to GL water density.

Figures 3–5 present the time changes of the water density fields recorded at three constant applied pressures of 225 bar (Fig. 3), 250 bar (Fig. 4), and 270 bar (Fig. 5), while increasing the temperature from 603 to 673 K, as indicated in Fig. 1b. Pressure is regulated by the backpressure valve (see Supplementary Fig. 3) and the pressure drop is considered negligible. Figures 3a–5a present a set of representative density maps. They were selected from the total number of ~250 images recorded during the experiments at representative moments in the timeline: when the color changes from blue to red, then the co-existence of orange reddish (LL) with yellow (GL), and then the conversion to pale yellow.

Figures 3b–5b present the time profiles of the temperature measured before the monolith. The main difference between the

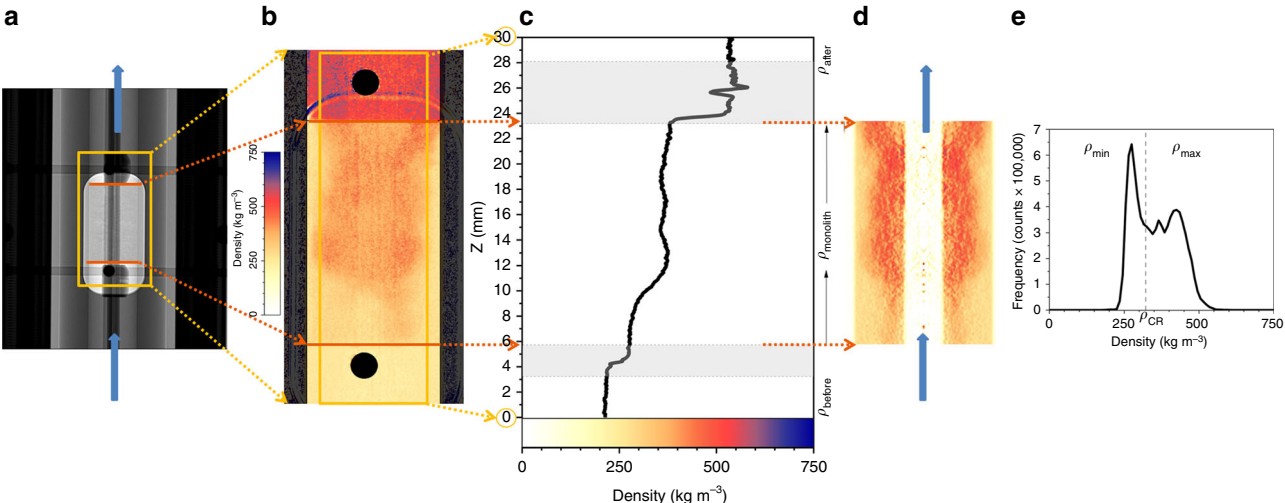

**Fig. 2** Example of the results presentation. **a** Neutron image of the monolith inside the continuous flow tubular reactor: the blue arrows indicate the water flow direction, the red lines indicate the monolith edges and the yellow square indicates the region of interest for images processing. **b** Color-scaled density map: LL supercritical water is reddish, GL supercritical water is yellowish and liquid water is blueish; the yellow square indicates the region of interest for the vertical density profile. **c** Water density profile: the three density regions of interest are marked by $\rho_{before}$, $\rho_{monolith}$, and $\rho_{after}$; the gray zones were not taken into consideration for density calculation, as they are image artifacts induced by the monolith edges and the reactor's window for neutrons. **d** Tomographic reconstruction of the monolith region and **e** Computed histogram showing the volume fraction of densities $\rho_{max}$ and $\rho_{min}$ within the monolith region. Source data for **c** and **e** are provided as a Source Data file

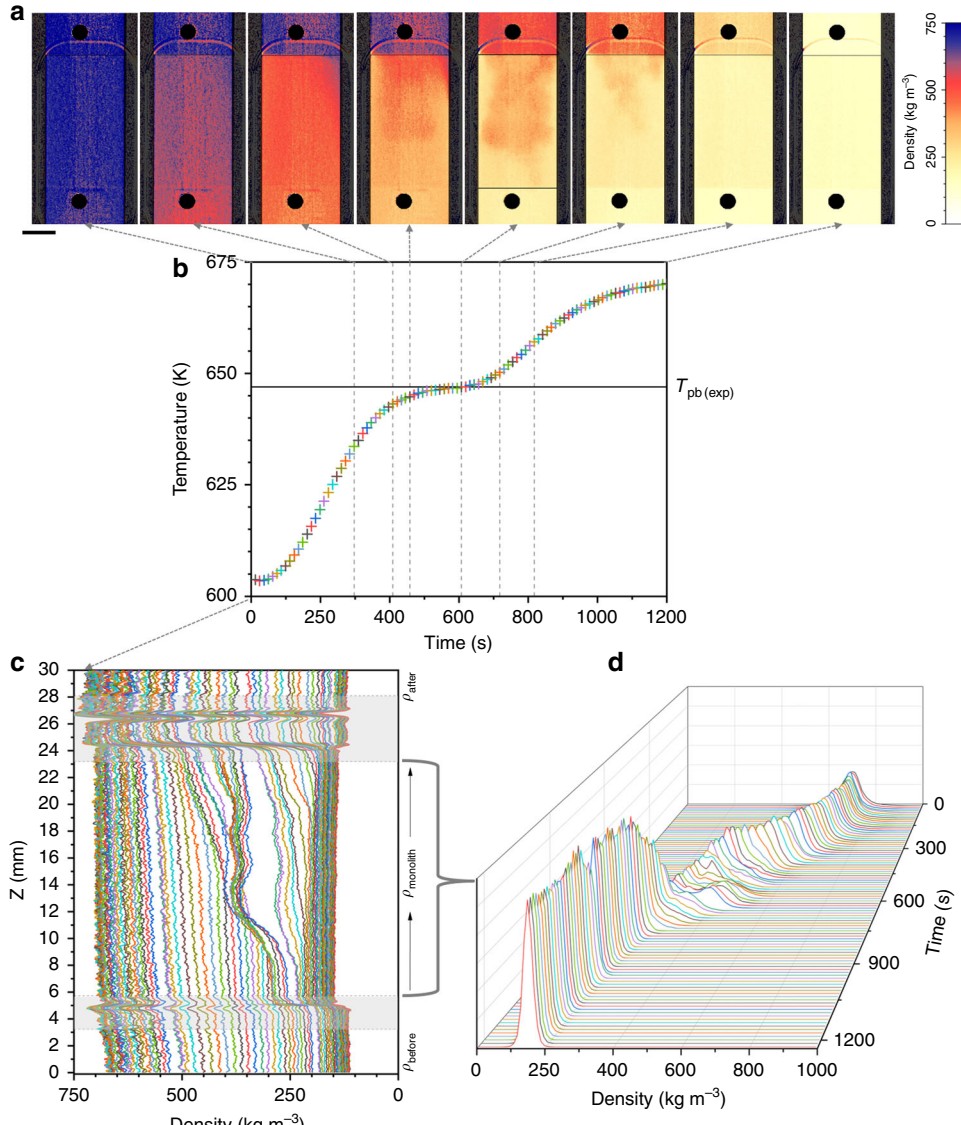

**Fig. 3** Time evolution of the system during isobaric heating at 225 bar. **a** A set of representative color scaled density maps are presented to indicate a few significant moments: when the color changes from blue to red, then the coexistence of orange reddish with yellow, and then the conversion to pale yellow; the length of the scale bar is 5 mm. **b** The temperature profile recorded before the monolith with the indication of the temperature of the plateau ($T_{pb(exp)}$). **c** Vertical densities profiles and **d** densities histograms for the monolith region. Each point in the temperature profile corresponds to a line in the density profile and a density histogram. Source data for **b**–**d** are provided as a Source Data file

three isobars is that the system reaches faster the final temperature at higher pressure. Independently from the pressure, there is a plateau observed in the temperature profile. Its persistence in time is shorter as the pressure increases. We associated the middle of the plateau to the *pseudo*-boiling temperature determined from our experiments ($T_{pb(exp)}$). Table 1 presents the values of $T_{pb(exp)}$ for each isobar in comparison with the theoretical values determined from NIST data[13] and based on Banuti's theory of *pseudo*-boiling[19,25,26,33]. The *pseudo*-boiling temperature increases as the pressure increases and this is consistent with our experimental observations.

In Figs. 3c–5c, we present the time line of the density profiles and Figs. 3d–5d show the density histograms. There are common features to notice, independently of the pressure when analyzing the density profiles. For all three isobars $\rho_{before} < \rho_{monolith} < \rho_{after}$, and the difference between the three regions increases as temperature increases. However, when the system reached the

final temperature at 225 bar, $\rho_{before}$ became equal to $\rho_{after}$ but lower than $\rho_{monolith}$. At the beginning of the time line of all three isobars, $\rho_{before}$ has a slightly ascendant slope towards the entrance of the monolith, while $\rho_{monolith}$ has a constant density value (flat profile). At a certain temperature in the time line, $\rho_{monolith}$ starts developing a profile monotonically increasing towards the exit of the monolith. In addition, on all time lines and independent of the temperature, $\rho_{after}$ has a flat profile showing a constant density.

There are three times of interest in Figs. 3–5, which corresponds to the times defining the plateau in the temperature profile. At the time when $\rho_{before}$ becomes constant, $\rho_{monolith}$ starts presenting the wave, which corresponds to the two peaks splitting in the density histogram. The intensity of this wave increases up to the time when the system before the monolith equals $T_{pb(exp)}$ and decreases afterwards until it disappears at the time when the density histogram starts to present again only one peak. To recall

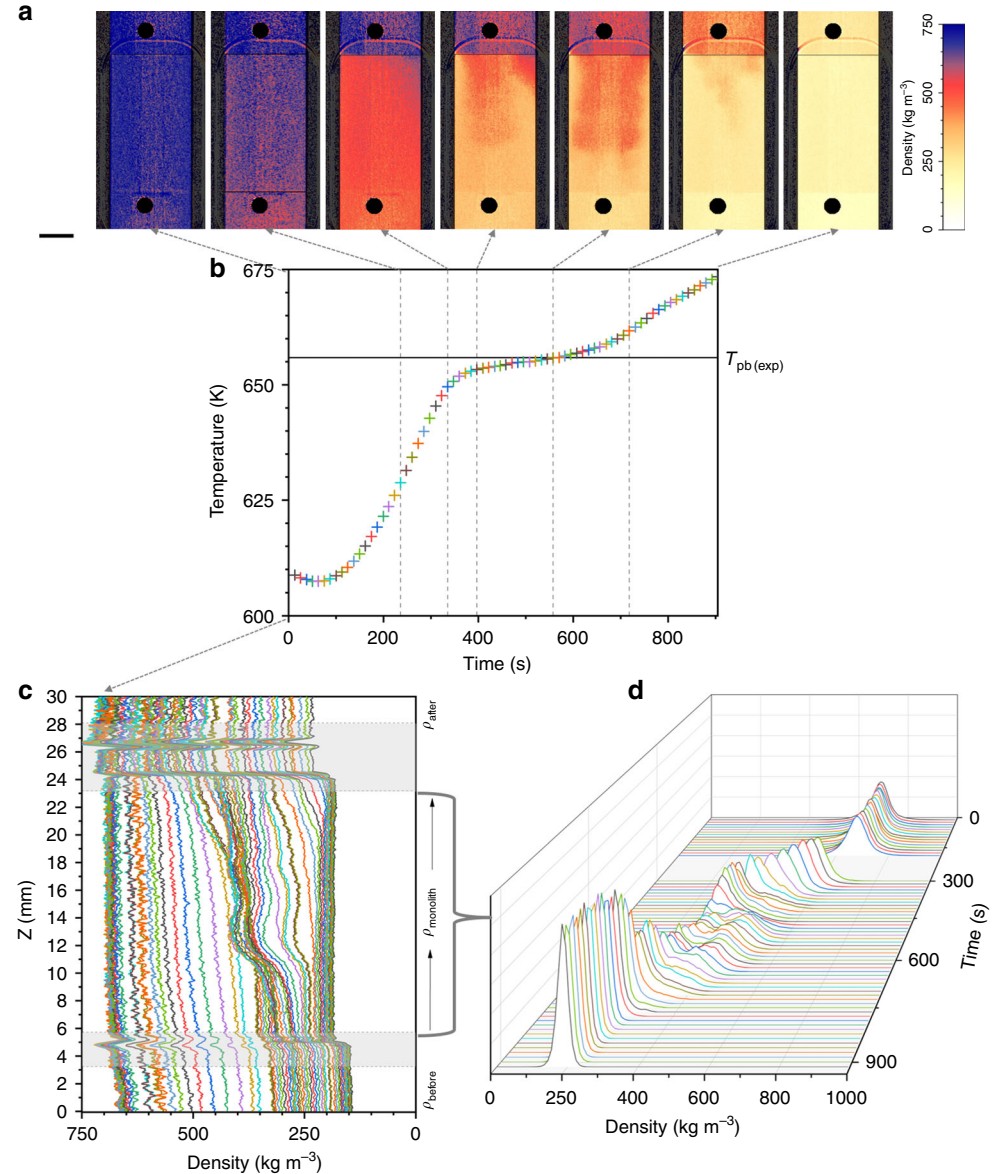

**Fig. 4** Time evolution of the system during isobaric heating at 250 bar. **a** A set of representative color scaled density maps are presented to indicate a few significant moments: when the color changes from blue to red, then the coexistence of orange reddish with yellow, and then the conversion to pale yellow; the length of the scale bar is 5 mm. **b** The temperature profile recorded before the monolith with the indication of the temperature of the plateau ($T_{pb(exp)}$). **c** Vertical densities profiles and **d** densities histograms for the monolith region. Each point in the temperature profile corresponds to a line in the density profile and a density histogram. Source data for **b**–**d** are provided as a Source Data file

from Fig. 2b, d, in the density maps this wave marks the volume of the monolith where the co-existence of LL (red) and GL (yellow) supercritical water is clearly visible.

The video clips including all the density images and presented as Supplementary Movies show fast fluctuations and random changes in the distribution of colors when the temperature increases, particularly during the time of the plateau in the temperature profile. Only a few of them could be displayed in Figs. 3a–5a. The complete spread of pale yellow over the entire field of view is noticed only at the end of the timeline at 225 bar.

**Density fluctuations outside monolith and monolith-free case.** Figure 6 presents the temperature variation of $\rho_{before}$ and $\rho_{after}$ using the density values extracted from the density profiles presented in Figs. 3c–5c. The bulk (free) water density values reported in the NIST database[13] are included in the graphs for

comparison. First observation from Fig. 6 is that, independently from the pressure, there is a difference between $\rho_{before}$ and $\rho_{after}$, with the water density after always higher than the density before the monolith. The two sigmoidal curves merge only at 225 bar when the system reached the final temperature. The merging is almost complete at 250 bar and incomplete at 270 bar. Secondly, the NIST density values ($\rho_{NIST}$) are in better agreement with $\rho_{before}$ than with $\rho_{after}$ and always $\rho_{before} < \rho_{NIST} < \rho_{after}$. The temperature dependence of both $\rho_{before}$ and $\rho_{after}$ follows the trend seen for $\rho_{NIST}$[13]. The density decay in the region of the Widom line associated to *pseudo*-boiling phenomenon[22,34] is smoother at higher pressures. Moreover, the higher the pressure, the larger is the difference between $\rho_{before}$ and $\rho_{after}$ and the maximum difference between the two is recorded at the temperature of *pseudo*-boiling irrespective of pressure.

Figure 6b includes the density variation recorded from experiments without monolith at 250 bar, and it can be noticed

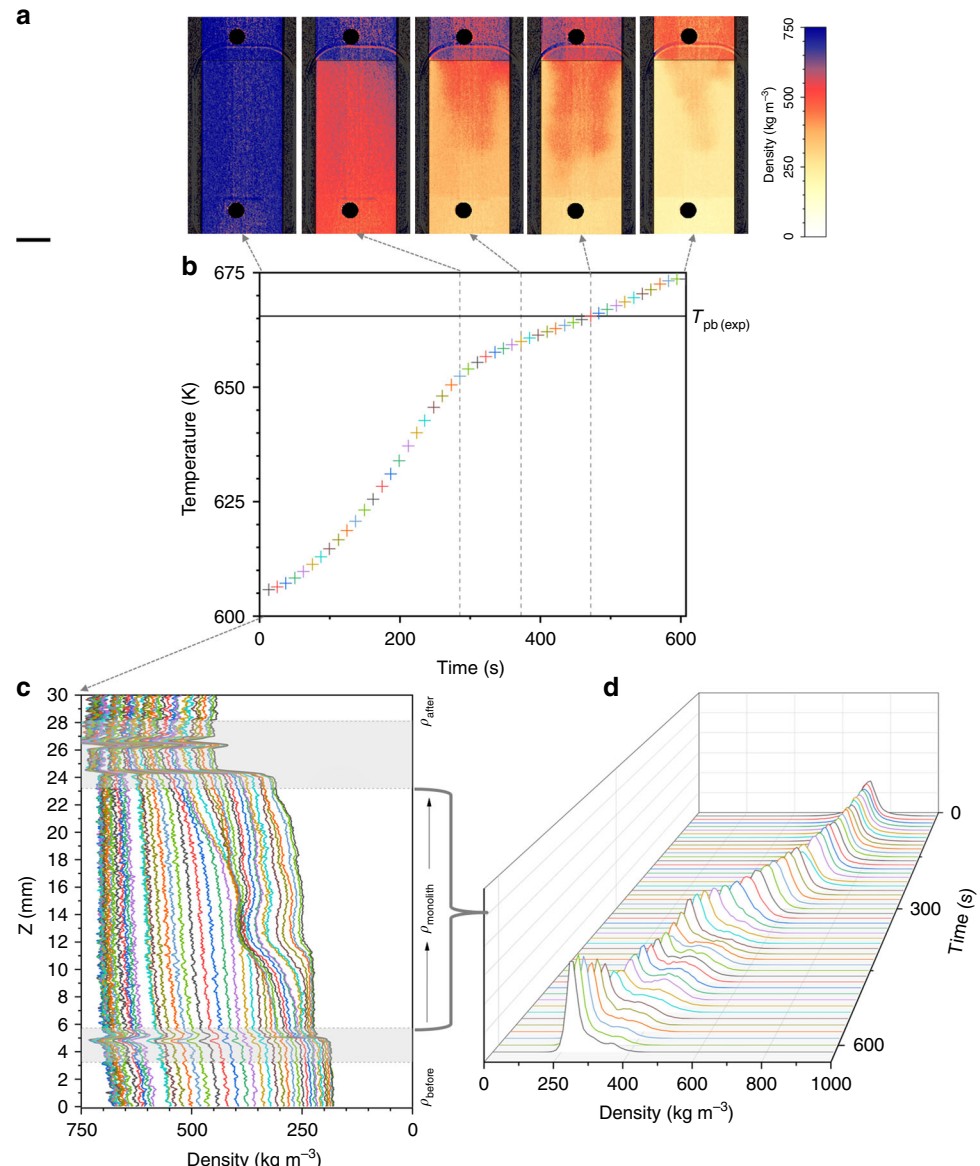

**Fig. 5** Time evolution of the system during isobaric heating at 270 bar. **a** A set of representative color scaled density maps are presented to indicate a few significant moments: when the color changes from blue to red, then the coexistence of orange reddish with yellow, and then the conversion to pale yellow; the length of the scale bar is 5 mm. **b** The temperature profile recorded before the monolith with the indication of the temperature of the plateau ($T_{pb(exp)}$). **c** Vertical densities profiles and **d** densities histograms for the monolith region. Each point in the temperature profile corresponds to a line in the density profile and a density histogram. Source data for **b**–**d** are provided as a Source Data file

**Table 1 The values of the *pseudo*-boiling temperature determined experimentally for each isobar $T_{pb(exp)}$**

| $P$ (bar) | $T_{pb(exp)}$ (K)[a] | $T_{WL\text{-}CP(max)}$ (K)[b] | $T_{pb(NIST)}$ (K)[b] |
|---|---|---|---|
| 225 | 646.9 | 648.7 | 649.0 |
| 250 | 655.6 | 658.1 | 659.0 |
| 270 | 664.9 | 665.2 | 667.0 |

[a]The accuracy of the temperature measurement is ±1.5 K (see the "Methods" section)
[b]The temperatures corresponding to the Widom line of maxima isobaric heat capacity[13,26] and the *pseudo*-boiling temperatures according to Banuti's theory[25]

that $\rho_{monolith\text{-}free}$ is in very good agreement with the theoretical curve of $\rho_{NIST}$.

Figure 7 shows the comparison of the water density images recorded at different temperatures during the isobaric heating at 250 bar in the experiments without (Fig. 7a) and with monolith (Fig. 7b). It can be easily noticed that without monolith the density fields are homogenous. Non-homogenous density distribution is recorded only in the presence of the monolith. Moreover, co-existence of LL and GL supercritical water inside the monolith is visible at the temperatures close to the $T_{pb(exp)}$.

**Density fluctuations inside monolith.** Figure 8 presents the timeline of the differences between $\rho_{min}$ and $\rho_{max}$ computed from the density histograms presented in Figs. 3d–5d (see the "Methods" section for details). The curve corresponding to the minimal density values is yellow and the one for the maximal values is red, to follow the colored-scale for the monolith region in the density images. The curves showing the density variation before and after the monolith are included for comparison. First, we observe that $\rho_{min}$ curve is nearly the same as $\rho_{before}$,

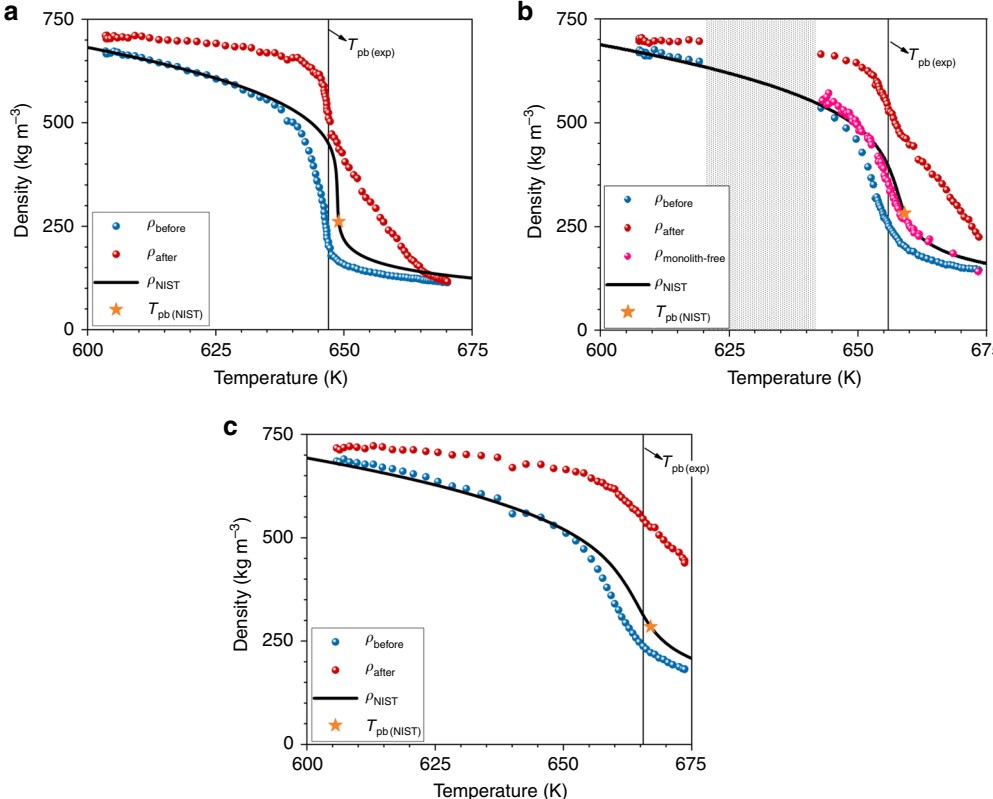

**Fig. 6** Temperature variation of water density outside monolith. It presents density variations recorded before ($\rho_{before}$) and after ($\rho_{after}$) the monolith at **a** 225 bar, **b** 250 bar, and **c** 270 bar; the NIST[13] reference values for bulk water density are included for comparison; the *pseudo*-boiling temperature $T_{pb(exp)}$ determined experimentally is indicated by the solid line; the orange symbol indicates the theoretical values of the temperatures corresponding to the *pseudo*-boiling line; for the 250 isobar the density variation with temperature for the monolith-free case are also presented ($\rho_{monolith\text{-}free}$). Source data are provided as a Source Data file

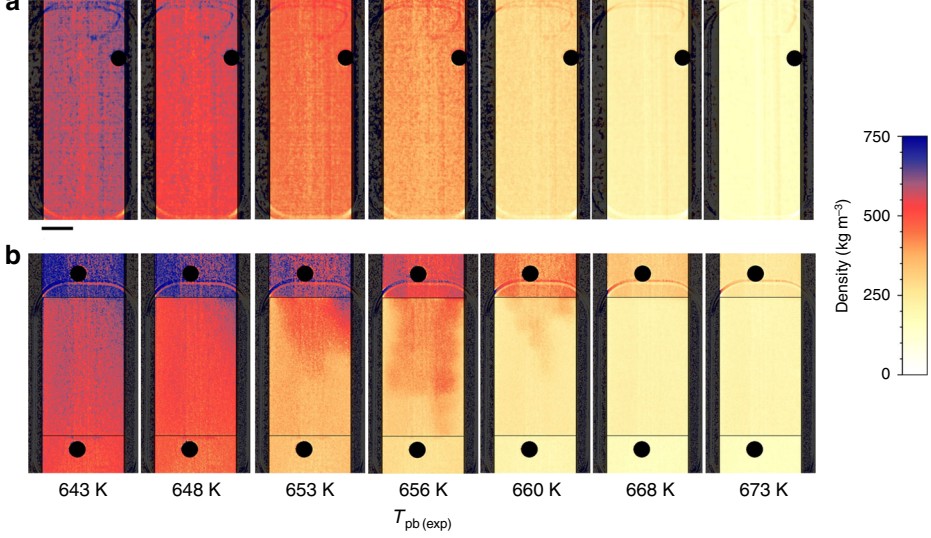

**Fig. 7** Comparison between monolith-free and monolith cases. It shows a set of color scaled density images at different temperatures recorded during isobaric heating at 250 bar in experiments without **a** and with **b** the monolith; it is evident that the density field is homogenous without monolith; the co-existence of LL and GL supercritical water is visible only in the presence of the monolith; the length of the scale bar is 5 mm

while the higher density in the monolith ($\rho_{max}$) is not as dense as the fluid after the monolith. Moreover, the yellow and red curves start to separate at the time when $\rho_{before}$ has the inflection point. We then observe the maximum separation of the curves at the time of *pseudo*-boiling, the time when the system before monolith equals $T_{pb(exp)}$. The two curves come back together at the time when $\rho_{after}$ has the inflection point. These observations are valid for all isobars.

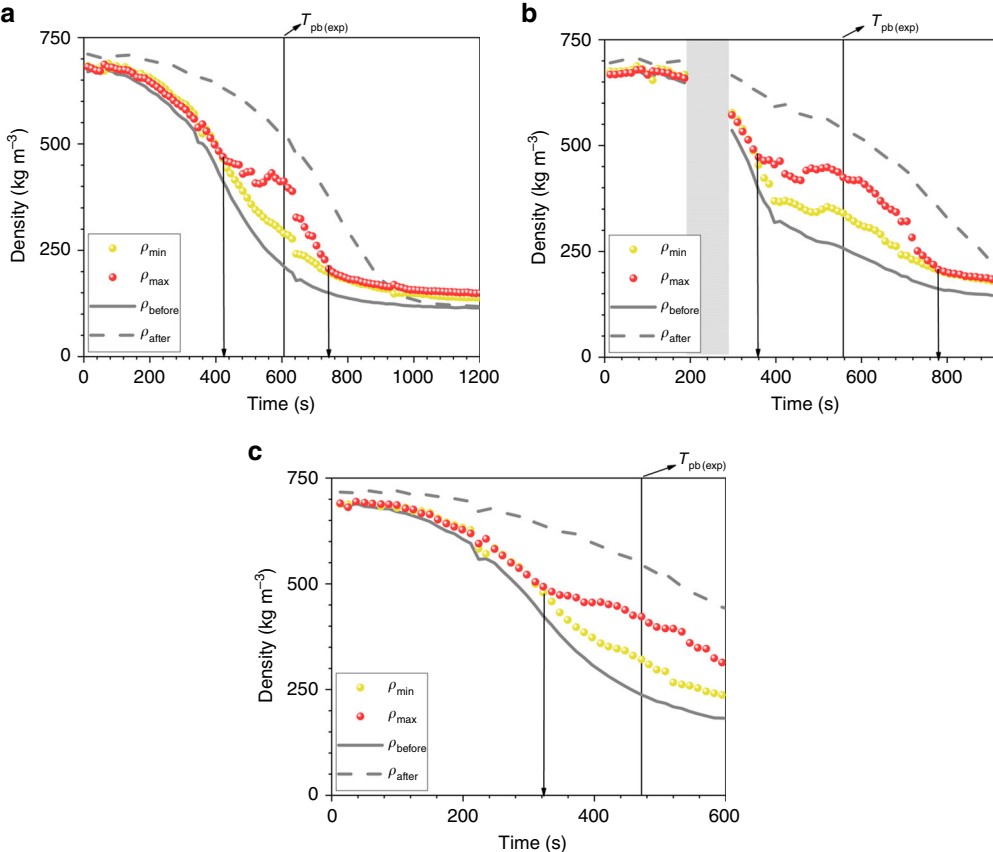

**Fig. 8** Time-domain analysis of water density recorded inside the monolith. It presents the variation of min and max densities ($\rho_{min}$ and $\rho_{max}$) at **a** 225 bar, **b** 250 bar, and **c** 270 bar; vertical lines indicate the times at which the curves starts to separate, reach the maximum separation at *pseudo*-boiling point and when they come back together; the curves showing the density variation before and after the monolith are included for comparison. Source data are provided as a Source Data file

## Discussion

At the scale of our experiments, which is the millimeters scale, water appears as a single-phase flowing upwards through the tubular reactor. In the presence of the monolith, supercritical water reveals its two-phase flow molecular features and we visualize the co-existence of LL and GL states. Our results clearly show that the fluid–solid interaction plays a fundamental role in the visualization of LL to GL supercritical water dynamic phase transition.

Supercritical *pseudo*-boiling is the concept used to explain the LL–GL transition at Widom line crossover[26]. According to *pseudo*-boiling theory, the system uses the energy provided for structural changes and for raising the temperature of the fluid during the transition. Therefore, there are two contributions, the structural and the thermal[26]. The two contributions can be distinguished only below the reduced pressure ($P_r = P/P_{CP}$) of 1.5 when the structural contribution exceeds the thermal one[26]. It is reported that the supercritical *pseudo*-boiling is detectable up to a pressure three times higher than the critical pressure[26,33]. We performed all our experiments below a reduced pressure of 1.5 (Fig. 1) and monitored the density fluctuations during the *pseudo*-boiling by neutron imaging (Fig. 2).

First indication that we detect the *pseudo*-boiling is the plateau in the temperature profile, temperature being recorded before the monolith (Figs. 3–5). In this region, the LL to GL phase transition has the same hydrodynamic conditions as in the monolith-free case. It is similar to subcritical boiling, when temperature remains constant until all liquid transforms into gas due to the latent heat

of vaporization. The plateau in our temperature profile is then indicating the time of the structural changes.

It is evident that the monolith brings different hydrodynamic conditions for the LL to GL supercritical water phase transition. The conversion from LL to GL supercritical water is delayed inside the monolith, as indicated by the dynamic lag between the water densities recorded before and after the monolith (Fig. 6). Moreover, when the *pseudo*-boiling is detectable in our experiments, we visualize the co-existence of LL and GL supercritical water only in the presence of the monolith (Fig. 7).

The monolith is a highly permeable carbon fibers porous material with hydrophobic surfaces. In our experiments, there is huge hydrophobic surface in contact with water inside the monolith. Hydrophobic surface is repulsive for LL water, which is polar[14], and clustering of water molecules via hydrogen bonding is likely to occur. The coalescence (attraction) of these clusters when water exits the monolith further increases the density in the absence of the hydrophobic surface and this is a possible reason why in the after monolith region we observe higher densities. There is a difference in hydrogen bonding index between LL and GL water, which changes the polarity and water in GL state becomes non-polar[14]. The hydrophobic surface is attractive for GL water. The competition between repulsive for LL and attractive for GL dominated water–hydrophobic surface interaction might be the reason of GL fraction separation inside the monolith at the time of *pseudo*-boiling (Fig. 8). The interplay between repulsion and attraction-dominated interactions might induce hysteresis behaviors as the ones we observed,

independently of pressure, in the experiments with the monolith. However, isobaric *pseudo*-boiling requires energy for both thermal and structural contributions[26], therefore local temperature reduction can also explain the density changes monitored during heating up the monolith at a constant pressure.

In the volume of the monolith, there is water confined in micropores and free water in the large interfibrillar voids. In the micropores, water stays packed to the maximum local density imposed by the strong field of confining adsorption forces. Small-angle X-ray scattering measurements at saturation capacity of hydrophobic micropores in activated carbon fibers estimate a density of 810–860 kg m$^{-3}$ [35]. The water in micropores maintains this density as long as LL water fills the interfibrillar volume. During the isobar heating, *pseudo*-boiling initiates when the temperature reaches the $T_{pb}$ values corresponding to constant applied pressures in each experiment (see Supplementary Fig. 5). As the *pseudo*-boiling line is reached, conversion of interfibrillar water from LL to GL causes desorption of water from the micropores. A brief calculation based on the Dubinin–Astakhov micropore-filling model[36,37] predicts that the density in micropores decays at constant external pressure as the temperature continues to increase above $T_{pb}$ (see Supplementary Fig. 6). The information about the amount of water within the micropores supports the claim that the large variations of water density revealed by neutron imaging are not caused by major density changes in micropore-confined water. In this study, micropores played a minor role in the observed density variations, and only above the *pseudo*-boiling temperature.

The density changes revealed by neutron imaging occur in the interfibrillar space, which represents close to 80% of monolith's volume. The *pseudo*-boiling occurs in the pools created by the interfibrillar voids. What video clips of our experiments actually show when water inside the monolith crosses the Widom line is the travel of the supercritical water *pseudo*-boiling front in the interfibrillar voids space. The voids between fibers are large enough to bring the front trajectory visible from the micrometers scale to the millimeters scale of our experiments.

In conclusion, with the help of neutron-imaging technique, we visualized the conversion from LL to GL supercritical water at Widom line crossover. We explained the experimental observations mainly based on supercritical water *pseudo*-boiling concept. The porous carbon monolith delays water structural changes, and thus the LL–GL two-phase flow instability became visible at millimeter scale. The monolith brings the time scale of the LL/GL transition at the accessible time scale of our experiments and it is practically helping to distinguish the structural contribution of the *pseudo*-boiling.

Our findings clearly show that LL and GL are two distinguishable supercritical water states at millimeter scale. Understanding the structure and the properties of these two distinct supercritical water metastable phases is relevant for establishing the optimal operating conditions of supercritical water reactors used in different technologies including thermochemical conversion of biomass into biofuel[2,38]. Moreover, our neutron-imaging method allows studying the water–solid surface interaction under supercritical conditions. This information is relevant for the design of materials used as catalysts, sorbents, or membranes for separation processes. For instance, the porous material with hydrophobic surface used in this study might have similar functionality as the recently developed membrane distillation for desalination[39]. Salts separation under supercritical water conditions is a relevant process for nutrients recovery during the biofuel production from biomass[40].

## Methods

**Neutron imaging experimental procedure**. For the neutron imaging experiments, ultrapure MiliQ degassed water was used as fluid and carbon fibers carbon molecular sieves sample was used as solid porous medium.

The carbon sample is a cylindrical monolithic body with a volume of around 2.15 cm$^3$ (see Supplementary Fig. 1a). Prior to the neutron radiography experiments, the monolith was characterized by SEM, nitrogen adsorption, and high-resolution X-ray diffraction (HRXRD). In short, the monolith is made of activated carbon fibers (ACF) with small diameter of about 17 μm in size. The fibers are not distinguishable in the neutron images due to too large pixel size; however, the fibers can be visualized by SEM (see Supplementary Figs. 1b, d). The material has low density (223 kg m$^{-3}$) and high permeability derived from the existence of large interfibrillar voids. The peculiar property is the high surface area (1360 m$^2$ g$^{-1}$) from a large number of narrow, internal pores in ACF. The N$_2$ adsorption characterization at 77 K showed a total pore volume of 0.62 cm$^3$ g$^{-1}$, of which 92% or 0.57 cm$^3$ g$^{-1}$ is located in micropores (pores with width $w < 2$ nm). The latter have bimodal pore distribution with maxima at 0.5–0.6 and at 1.0–1.2 nm (see Supplementary Fig. 2). After the experiment under supercritical conditions, the monolith sample showed microstructural and structural stability determined by SEM and HR-XRD, respectively (see Supplementary Fig. 1b, c). This agrees with literature reports that supercritical water does not affect the pore structure of carbon fibers monoliths at temperatures below 973 K[41].

The neutron imaging experiments were performed using a dedicated setup for in situ measurements under supercritical water conditions (designed and developed by SITEC-Sieber Engineering AG, Switzerland and Computer Power SRL, Romania). The main component of the setup is the continuous flow vertical tubular reactor with an inner diameter of 12 mm and made of Zircaloy-4, material suitable for high pressure–high temperature experiments. The optimum reactor wall thickness was determined by taking into account both the maximum operating conditions ($T_{max} = 673$ K and $P_{max} = 300$ bar) and the neutron attenuation coefficient of Zircaloy 4. The schematic representation of the setup is presented in Supplementary Fig. 3.

As mentioned in the main text, the water was supplied continuously into the reactor, which contains the monolith, at a constant flow rate of 5 mL min$^{-1}$ using a high precision liquid chromatography (HPLC) pump. The direction of the water flow through the monolith is upwards.

Initially, the monolith was flooded with water and pressure was increased to 270 bar using a backpressure regulator. Along the setup's line, there are two measuring points for the pressure: at the inlet using the HPLC pump sensor, and at the outlet using pressure sensor. It is worth to mention here that the pressure sensors are placed in the zones where water is at room temperature. Although the setup was designed to maintain constant pressure in the reactor, localized pressure variations in the zone of the monolith could not be recorded.

After applying the pressure, the water was first heated to 573 K in the embedded preheater placed at the inlet. Inside the reactor, the water was uniformly heated up to 673 K using the aluminum block heater, which has been designed especially for optimum heat transfer to the reactor wall. Right downstream of the reactor, the water was cooled down to room temperature using a heat exchanger (cooler). The temperature was measured in four points using thermocouples placed in the preheater, the heater, inside the reactor, and at the cooler's exit. It is to mention here that the reactor thermocouple support was used to fix the monolith sample. Moreover, the temperature measuring point inside the reactor is at 30 mm beneath the monolith and the accuracy of the measurement is ±1.5 K.

The flow rate, the pressure, and the temperature were monitored and recorded with a step of 10 s/data reading.

Three sets of measurements corresponding to the three isobars were performed. The pressure was set at 225, 250, and 270 bar and the reactor temperature was increased from 603 to 673 K.

Neutron images were acquired during heating for the three isobars. The measurements including the carbon porous monolith were performed at the ICON beam line[42] of the Paul Scherrer Institute (PSI), which has a cold neutron energy spectrum covering a wavelength range from 1 to 8 Å with an intensity weighted average wavelength of 3.1 Å. The flux of the neutron beam was ~1.5 × 10$^7$ n cm$^{-2}$ s$^{-1}$. The neutrons were captured and converted to visible light using a 20 μm-thick scintillator screen made of Gd$_2$O$_2$S and the resulting image was recorded by a CCD camera (Andor Ikon-L, 2048 × 2048 pix). Each image was recorded with an exposure time of 10 s. The background contribution due to scattered neutrons was measured using our in house developed method[43] based on a black body grid. The grid consisted of black bodies made of $^{10}$B$_4$C (a strong neutron absorber) of cylindrical shape, with a diameter of 2.5 mm and a length of 3 mm in the beam direction. They were held together in a square grid arrangement using an aluminum frame, with a center-to-center distance of 25 mm. The measurement without the carbon sample were performed at the NEUTRA beam line[44], which has an energy spectrum in the thermal range with an intensity-weighted average wavelength of 1.8 Å. The flux of the NEUTRA beam was ~0.6 × 10$^7$ n cm$^{-2}$ s$^{-1}$. In this case, a 50 μm-thick scintillator screen made of $^6$Li/ZnS was used to convert the neutrons and the resulting visible image was captured with a scientific CMOS camera (Andor Neo sCMOS, 2160 × 2560 pix). Each image was recorded with an exposure time of 30 s, and the same scattered background correction procedure was used.

**Image processing**. The raw images corrected for the camera offset, were filtered to remove the white spots corresponding to gamma rays hitting the detector and the noise in high spatial frequencies, and corrected for the background contributions due to scattered neutrons. The latter step was performed by generating a background image by interpolating the intensity measured behind the black bodies, and subtracting this background from the acquired image. Then, the image was corrected for the beam intensity fluctuations based on the intensity measured in a known area of constant transmission, and for the local detector sensitivity based on an image of the open beam. After this, it was divided pixel-wise by the image of the dry reactor to remove the contributions from the attenuation of the reactor itself. The corresponding relative transmission image $\left(\frac{I}{I_0}\right)$ was converted to an equivalent water thickness image ($\delta$) using the following equation:

$$\delta = -k_1 \ln\left(\frac{I}{I_0}\right) - k_2 \left[\ln\left(\frac{I}{I_0}\right)\right]^2 - k_3 \left[\ln\left(\frac{I}{I_0}\right)\right]^3 \tag{1}$$

The non-linear relation between $\delta$ and $\ln\left(\frac{I}{I_0}\right)$ stems from beam hardening (a reduction of the effective neutron cross section with increasing material thickness due to the energy dependency of the neutron cross section). For the experiments performed at ICON, the values of the parameters $k_1$, $k_2$, and $k_3$ were determined using the ICON beam spectrum and the energy-dependent cross section of water, and detection efficiency. For water at room temperature, known cross section were used[45] resulting in the following values: $k_1 = 2.19$ mm, $k_2 = 0.062$ mm, and $k_3 = 0.0012$ mm. For water at higher temperatures (600–700 K), the energy-dependent cross sections were measured at ICON using the reactor described above, nevertheless without water inside. The resulting values where $k_1 = 1.87$ mm, $k_2 = 0.16$ mm, and $k_3 = -0.0055$ mm. The equivalent thickness image was finally converted to a density image by dividing pixel-wise the thickness with the measured thickness in a reference condition, where the reactor was filled with liquid water at room temperature and with a pressure of 270 bar. For display purposes, the images were converted in false color with blue representing the liquid water density, red representing the density of LL supercritical water ($\rho$ around 600 kg m$^{-3}$) and yellow representing the density of GL supercritical water ($\rho$ around 200 kg m$^{-3}$).

For the experiment at 250 bar, we discarded seven images due to failed acquisition. Therefore, water density data for these images were not included, and that region is marked with gray in the graphs.

For the experiments performed at NEUTRA, the same procedure was used, but the room temperature liquid water reference was acquired without pressurizing. In that case, the parameter values for Eq. (1) were the following: $k_1 = 2.79$ mm, $k_2 = 0.066$, $k_3 = 0.0009$ mm for room temperature water and $k_1 = 2.73$ mm, $k_2 = 0.1$ mm, $k_3 = -0.002$ mm for high-temperature water.

Vertical density profiles were obtained by averaging all density values on the same horizontal line, discarding the pixels at the black body locations and outside the reactor.

**Calculation of min/max densities**. Due to the fact that each line in the direction of the neutron beam can contain different density values, the minimal and maximal values obtained in the projected image are not necessarily the true minima and maxima. To correct for this, a tomographic reconstruction was performed based on the assumption of axial symmetry, using the onion peeling method[46]. For each resulting image, a histogram of density was computed, weighting the pixels according to their distance to the symmetry axis to take into account the volume they represent in the 3D space. Finally, the minimum and maximum values were computed by using the 5% and 95% percentiles based on the histogram, and correcting them for the histogram peak broadening due to image noise. The difference between the 5% and 95% percentiles was observed to be approximately proportional to the average value in the images having a homogeneous density distribution. As a consequence, the maximal density was estimated by multiplying the 95% percentile by a factor 0.87 and the minimal value by multiplying the 5% percentile by a factor 1.13. These correction factors were chosen so that the minimal and maximal values match in situation assumed homogeneous, such as the initial and final images of the 225 bar experiment.

**Measurements of water adsorption on activated carbon fibers**. Water adsorption and desorption isotherms were measured on activated carbon fibers representative for the composition of the monolith. Measurements were done using the dynamic vapor sorption (DVS resolution) instrument from Surface Measurements Systems. Measurements were carried out at 25, 35, and 55 °C over the range of relative pressures $0 < P/P_0 < 0.95$. Data were fitted with the Dubinin–Astakhov (DA) equation:

$$W(P) = W_0 \left[ -\left(\frac{A}{E}\right)^n \right] \tag{2}$$

where $W(P)$ is the amount adsorbed (g/g) at the equilibrium pressure $P$, $W_0$ is the maximum capacity filling of the micropore volume at the saturated vapor pressure $P_0$ and temperature $T$, $E$ is the characteristic energy for the adsorbent–adsorbate pair, $n$ is a structural homogeneity parameter and $A$ is the adsorption potential, $A = RT \ln(P_0/P)$[36,37,47]. When plotted versus the adsorption

potential, all experimental data at three temperatures collapse over a unique adsorption isotherm, which characterizes water adsorption on the activated carbon fibers of the experiment (see Supplementary Fig. 4). Parameters found by fitting ($E = 1.3$ kJ mol$^{-1}$ and $n = 2.3$) are in the range expected for water adsorption in microporous carbons[36,37]. The maximum micropore capacity ($W_0 = 0.45$ g$_{H_2O}$/g$_{carbon}$) was calculated from the micropore volume obtained from nitrogen adsorption (0.53 cm$^3$/g$_{carbon}$) and the highest density of tightly packed $H_2O$ in carbon micropores (0.86 g$_{H_2O}$/cm$^3$).

## Data availability
The raw data acquired for these experiments can be obtained from the authors upon reasonable request. The source data underlying Fig. 1a, b, 2c, e, 3b–d, 4b–d, 5b–d, 6a–c, and 8a–c and Supplementary Figs. 1c, d, 2, and 4–6 are provided as Source Data file with the paper. The reference data used in Figs. 1 and 6 have been obtained from publicly available NISTChemistry WebBook for "Thermophysical Properties of Fluid Systems".

## Code availability
Raw data were generated at the SINQ-PSI, Switzerland large-scale facility. The code used for images processing and derived data supporting the findings of this study are available from the corresponding author upon request.

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

## Acknowledgements
The authors acknowledge the financial support via PSI-CROSS-2016 funding and the Swiss National Science Foundation (project no. 153314). Cristian Iacob and Stefan Nicolae from Computer Power SRL, Romania are truly acknowledged for their work on building the setup and for technical assistance during the experiments. We thank Albert Schuler, Marcel Hottiger, Petter Vontobel and Jan Hovind for their technical support during different project phases and Dr. Eberhard Lehman, Dr. Rudolf Struis, and Prof. Frédéric Vogel for valuable discussions during the early stage of this project.

## Author contributions
F.M., K.K. and P.B. performed the experiments and carried out the data analysis. C.C. provided the monolith samples, and analyzed new water adsorption and desorption measurements. A.T. contributed to the manuscript revision. B.N. and C.L. are the co-leaders of the project. All authors contributed to the discussion of the results and to the manuscript writing.

## Additional information

**Competing interests:** The authors declare no competing interests.

