## [Peer Review File · Nature Communications]

Reviewers' comments:

Reviewer #1 (Remarks to the Author):

In this manuscript, Maxim et al. apply neutron imaging techniques to study the behavior of water in the supercritical region, in the proximity of the liquid-gas critical point (LGCP). Theory and computer simulations indicate that in the supercritical region, there are lines of maxima in thermodynamic response functions that originate at the LGCP, such as isobaric heat capacitance and isothermal compressibility. These lines converge on a single line as the LG critical point is approached, defining the so-called Widom line. At the Widom line, the correlation length of the system is maximum. In particular, the system is liquid-like at temperatures below the Widom line and it is gas-like at temperatures above the Widom line. A Widom line has been identified in experiments of Argon [21] but it has not been reported in experimental studies of water. In this work, the authors provide evidence that a Widom line can be identified for the case of water and that the system evolves rapidly from liquid-like (LL) to gas-like (GL) as the Widom line is crossed during isobaric heating. It is shown that the LL and LG states can be identified macroscopically, from the reported density field images.

The manuscript addresses an important topic since, as they explain, there are fundamental questions in the field of supercritical water that remain open. In addition, supercritical water has many potential applications, such as water treatment and in the conversion of biomass into fuel (the authors provide examples of applications in their intro). In addition, the behavior of water in the supercritical region above the LGCP is relevant to understand the behavior of water in the "supercritical region" above the hypothesized liquid-liquid critical point. A Widom line has been observed in computer simulations [L. Xu et al., PNAS 2005, 102, 16558] above the liquid-liquid critical point, consistent with experiments [Science 2017, 358, 1589].

A weak point of this work is related to the interpretation of the results. The authors study the density of water as it flows through a porous material. The Widom line is a thermodynamic property so, if it is macroscopically detectable, one should be able to observe the associated LG-LL transformation in water, in the absence of flow and with no porous material. In my view, the complexity of the experiments may open some questions in the interpretation of the results. Accordingly, I do not feel comfortable supporting publication of this work in Nature Communications.

1) I believe the authors use a porous material because "porous media induce density fluctuations in fluids near a critical point that are caused by fluid confinement inside the pores". However, it is not clear how the confining material affects the results.

In addition, it remains unclear what the role of water confined within the carbon fibers is. This is important since the phase behavior of water confined at the nanoscale can be very different from the phase behavior of bulk water.

2) Could there be any pressure gradient through the sample? This could also induced a non-uniform density distribution.

3) What is the rationale to identify ρ_{LL} as the maximum density within the monolith region? Similarly, why is ρ_{LG} attributed to the minimum density in the density within the monolith region? (lines 150-152). This is not explained in the manuscript.

Reviewer #2 (Remarks to the Author):

This paper reports really novel results on the Widom lines of supercritical water, which has an general interest. Authors succeeded to get the direct experimental evidence on the presence of different phases of supercritical water with neutron scattering and porous carbons. Then, I recommend publication of this paper on Nature Communications.

I have several questions and comments for the better revised manuscript.

1. Authors use activated carbon fiber monolith composites. The apparent density is very small compared with that of graphite. Therefore, the composites have great inhomogeneity in the sample from micropores, mesopores, and macropores (and interparticle voids). Surely the large macropores enable rapid diffusion of water in the carbon monoliths, as authors mention. However, the pore size should be quite critical in the state of water even at high temperature. For example, water should stay in micropores longer than in macropore-channels. There is a possibility that water in micropores and in macropores are associated with LL and GL, respectively. Then, authors may analyze the average size of red and yellow regions of the visualized images and show no relationship between those region sizes and pore sizes. Probably the images come from the averaged states in the depth direction and thereby authors' conclusion should be correct. In order to strengthen the conclusion I recommend this analysis.

2. Role of activated carbon fiber monoliths may be clearly described. Authors mention that confinement of fluids gives unusual critical behavior in mesopores. The confinement effect in micropores is considerably different from that in mesopores, because the interaction of fluids with

micropores is much stronger than with mesopores. As authors use microporous carbons, the description on mesopores may be distinguished from that on micropores.

3. Nanoconfinement effect in micropores is too strong to vary the critical behavior and thereby the water in micropores may be different from bulk water even at very high temperature. The water molecule-carbon surface interaction is about - 7 kJ/mol, corresponding to 800 K. In case of the small micropores, the plausible attractive interaction corresponds to 2×800 K. Authors may add some note on this point. I assume that activated carbon monolith can realize highly concentrated water system which enables to give the clear neutron images. If this is true, authors may mention this clearly.

4. Expression of "Density fluctuations induced by carbon porous material" is not appropriate. The density fluctuations should come from the phase behavior above the critical point. I wonder if the carbon porous materials just give the supercritical water which can be detected clearly by neutron imaging.

5. Could you check the IUPAC recommendation on pore classification?

Pure Appl Chem. 2015. Ultramicropores; pore width < 0.7 nm,

Supermicropores; 2 nm > pore width >0.7 nm

6. Can you estimate density(GL) and density(LL) by use of the apparent density of activated carbon monolith and pore volumes of micropores and (mesopores) ?

You can estimate the adsorption amount of water in micropore. If water in the micropores is predominant, you could estimate the densities of LL and GL water.

This sounds important.

7. T1, T2, and T3 are not clearly shown in Fig. 3

8. Hysteresis behaviors in Fig.4 are interesting. A more clear explanation is preferable.

Point-by-point response

Reviewer #1

Reviewer's comment: In this manuscript, Maxim et al. apply neutron imaging techniques to study the behavior of water in the supercritical region, in the proximity of the liquid-gas critical point (LGCP). Theory and computer simulations indicate that in the supercritical region, there are lines of maxima in thermodynamic response functions that originate at the LGCP, such as isobaric heat capacitance and isothermal compressibility. These lines converge on a single line as the LG critical point is approached, defining the so-called Widom line. At the Widom line, the correlation length of the system is maximum. In particular, the system is liquid-like at temperatures below the Widom line and it is gas-like at temperatures above the Widom line. A Widom line has been identified in experiments of Argon [21] but it has not been reported in experimental studies of water. In this work, the authors provide evidence that a Widom line can be identified for the case of water and that the system evolves rapidly from liquid-like (LL) to gas-like (GL) as the Widom line is crossed during isobaric heating. It is shown that the LL and LG states can be identified macroscopically, from the reported density field images.

Authors' Response: Indeed, now it is generally accepted that in the fluid's supercritical region the Widom marks the change in behavior between LL and GL fluid states. Recently, the Widom line it was associated to the *pseudo*-boiling phenomenon [Banuti, J Supercrit. Fluids, 2015, 98, 12], and it has been demonstrated that it is the extension of the coexistence line of liquid-gas in the subcritical region [Banuti, Phys. Rev. E, 2017, 95].

Reviewer's comment: The manuscript addresses an important topic since, as they explain, there are fundamental questions in the field of supercritical water that remain open. In addition, supercritical water has many potential applications, such as water treatment and in the conversion of biomass into fuel (the authors provide examples of applications in their intro). In addition, the behavior of water in the supercritical region above the LGCP is relevant to understand the behavior of water in the "supercritical region" above the hypothesized liquid-liquid critical point. A Widom line has been observed in computer simulations [L. Xu et al., PNAS 2005, 102, 16558] above the liquid-liquid critical point, consistent with experiments [Science 2017, 358, 1589].

Authors' Response: Our results can contribute to the fundamental understanding of water behavior during the LL to GL phase transition at the Widom line crossover. Indeed, it can be found similarities with the Widom line related to the liquid-liquid phase transition.

Reviewer's comment: A weak point of this work is related to the interpretation of the results. The authors study the density of water as it flows through a porous material. The Widom line is a thermodynamic property so, if it is macroscopically detectable, one should be able to observe the associated LG-LL transformation in water, in the absence of flow and with no porous material. In my view, the complexity of the experiments may open some questions in the interpretation of the results. Accordingly, I do not feel comfortable supporting publication of this work in Nature Communications.

Authors' Response: We fully agree with the reviewer. In order to clarify these points for the reader we provide in the revised manuscript the evidence that the LL-GL transformation occurs also in free water (no monolith) and can be monitored by neutron imaging. However, that is a smooth and continuous transition. In presence of the monolith we observe distinct zones of LL and GL that co-exist. Adding the monolith favor spatial separation and visualization of metastable LL and GL zones.

Reviewer's comment: 1) I believe the authors use a porous material because "porous media induce density fluctuations in fluids near a critical point that are caused by fluid confinement inside the pores". However, it is not clear how the confining material affects the results. In addition, it remains unclear what the role of water confined within the carbon fibers is. This is important since the phase behavior of water confined at the nanoscale can be very different from the phase behavior of bulk water.

Authors' Response: With additional experiments of water adsorption at subcritical conditions on similar activated carbon fibers, we derived the Dubinin-Astakhov (DA) characteristic curve for this system. Using this information, we were able to determine the amounts of water adsorbed in carbon micropores at the conditions of the experiment. The characteristic curve obtained this way was further used to estimate the behavior of water confined in micropores in supercritical conditions, before and after the *pseudo*-boiling temperature at each externally applied pressure. To do that, we used the old Dubinin's concept of *pseudo*-saturation pressure of supercritical fluids, and identified it with the temperature-dependent pressure on the *pseudo*-boiling line, as defined by Banuti in several papers [Banuti, J Supercrit. Fluids, 2015, 98, 12; Banuti, Phys. Rev. E, 2017, 95]. Supplementary Information file provides this analysis. The results show that (1) the amount of water present in the micropores remains constant up the temperature where the pressure of supercritical water (based on supercritical water properties from the literature) becomes equal to the constant external pressure applied in each experiment, when the *pseudo*-boiling transition occurs; (2) above that temperature, the amount of water present in micropores decreases as predicted by the DA characteristic curve; (3) the micropores volume is about 12 % of the total monolith volume, and the remaining 78 % is interfibrillar volume where water behaves like free bulk water outside the monolith; (4) it is this interfibrillar water that is observed by neutron imaging, where the temporary co-existence of LL and GL phases is visualized. Based on the analysis presented above, we revised the interpretation of the results and we demonstrated that the phase behavior of water confined at nanoscale plays a very minor role in the density fluctuations we monitored by neutron imaging.

Reviewer's comment: 2) Could there be any pressure gradient through the sample? This could also induced a non-uniform density distribution.

Authors' Response: Our computational fluid dynamics simulations indicate that the pressure gradient through the porous carbon sample is in the range of some Pascal therefore it cannot explain the big difference between the water density recorded before and after the monolith. However, there might be a small pressure difference between the inlet and outlet after the *pseudo*-boiling conditions have been reached,

which prompt desorption of water immobilized in micropores and its expulsion in the interfibrillar space. This water may not convert instantaneously into GL water, and after conversion may contribute to increased pressure upstream of the monolith (compared with the monolith free equivalent conditions). However, as mentioned above, the amount of water in micropores is small compared with that of free water in the interfibrillar space.

Reviewer's comment: 3) What is the rationale to identify ρ_{LL} as the maximum density within the monolith region? Similarly, why is ρ_{LG} attributed to the minimum density in the density within the monolith region? (lines 150-152). This is not explained in the manuscript.

Authors' Response: In the experiments with the monolith, we observed that during the time lag of the thermodynamic transition, *i.e.* during *pseudo*-boiling, distinct red and yellow regions co-exist within the monolith volume. According to the colored density scale, red color indicates densities LL and yellow is for GL densities. Obviously, density of LL water is higher than the density of water in GL state, and this is the reason why we identify LL as the maximum density and GL with minimum. After more analysis of density data, we present in the revised manuscript the values of max and min densities inside the monolith estimated after the tomographic reconstruction of the density maps.

Reviewer #2

Reviewer's comment: This paper reports really novel results on the Widom lines of supercritical water, which has a general interest. Authors succeeded to get the direct experimental evidence on the presence of different phases of supercritical water with neutron scattering and porous carbons. Then, I recommend publication of this paper on Nature Communications.

Authors' Response: In the revised manuscript we provide additional proof that the phase separation is observed only in the presence of the carbon monolith. This confirms one more time the role of the carbon monolith.

Reviewer's comment: 1. Authors use activated carbon fiber monolith composites. The apparent density is very small compared with that of graphite. Therefore, the composites have great inhomogeneity in the sample from micropores, mesopores, and macropores (and interparticle voids). Surely the large macropores enable rapid diffusion of water in the carbon monoliths, as authors mention. However, the pore size should be quite critical in the state of water even at high temperature. For example, water should stay in micropores longer than in macropore-channels. There is a possibility that water in micropores and in macropores are associated with LL and GL, respectively. Then, authors may analyze the average size of red and yellow regions of the visualized images and show no relationship between those region sizes and pore sizes. Probably the images come from the averaged states in the depth direction and thereby authors' conclusion should be correct. In order to strengthen the conclusion I recommend this analysis.

Authors' Response: We totally agree with the reviewer. In the revised manuscript we provide additional neutron images showing density measurements in absence of the monolith. The comparison shows minimal overall difference at equivalent conditions, but the phase separation between LL and GL regions is visible only in the presence of the monolith. Moreover, the size of red and yellow regions is in no way correlated with the size of the fibers (a few microns) or that of micropores present in the fibers (nanometers). The resolution of neutron imaging is not high enough to allow visualization of individual carbon fibers. Therefore, the density maps represent averaged projection on a 2D plane of real distribution of density pockets in the 3D monolith. Based on the tomographic reconstruction we estimated the real distribution of red and yellow regions within the monolith region.

Reviewer's comment: 2. Role of activated carbon fiber monoliths may be clearly described. Authors mention that confinement of fluids gives unusual critical behavior in mesopores. The confinement effect in micropores is considerably different from that in mesopores, because the interaction of fluids with micropores is much stronger than with mesopores. As authors use microporous carbons, the description on mesopores may be distinguished from that on micropores.

Authors' Response: Carbon fibers contain micropores, which strongly adsorb and immobilize water. In the interfibrillar space, the separation between carbon fibers is so large (tens of microns) that these voids would not qualify as mesopore space in the sense of IUPAC definition and cannot have any role in water adsorption.

With additional experiments of water adsorption at subcritical conditions on similar activated carbon fibers, we derived the Dubinin-Astakhov (DA) characteristic curve for this system. Using this information, we were able to determine the amounts of water adsorbed in carbon micropores at the conditions of the experiment. The characteristic curve obtained this way was further used to estimate the behavior of water confined in micropores in supercritical conditions, before and after the *pseudo*-boiling temperature at each externally applied pressure. To do that, we used the old Dubinin's concept of *pseudo*-saturation pressure of supercritical fluids, and identified it with the temperature-dependent pressure on the *pseudo*-boiling line, as defined by Banuti in several papers [Banuti, J Supercrit. Fluids, 2015, 98, 12; Banuti, Phys. Rev. E, 2017, 95]. Supplementary Information file provides this analysis: "Analysis of water density in activated carbon fiber micropores". Based on the analysis we concluded that only the interfibrillar water phases are visualized by neutron imaging. The external surface of carbon fibers, however, might help separate LL and GL phases by differential interactions with LL water clusters and GL water molecules.

Reviewer's comment: 3. Nanoconfinement effect in micropores is too strong to vary the critical behavior and thereby the water in micropores may be different from bulk water even at very high temperature. The water molecule-carbon surface interaction is about - 7 kJ/mol, corresponding to 800 K. In case of the small micropores, the plausible attractive interaction corresponds to 2 x 800 K. Authors may add some note on this point. I assume that activated carbon monolith can realize highly concentrated water system which enables to give the clear neutron images. If this is true, authors may mention this clearly.

Authors' Response: We thank the reviewer for this very important suggestion. As introduced above, we have done additional measurements of water adsorption and desorption. After fitting the data with DA equation, we estimated that the characteristic energy for water molecule-carbon surface interaction is 1.3 kJ/mol, close to the value estimated by the reviewer. Moreover, we distinguish between water confined in the micropores at high densities (860 kg/m^3 based on literature reports) and free water in the interfibrillar space with variable density (between about 700 kg/m^3 for LL, and less than about 200 kg/m^3 for GL). When the contribution of water density adsorbed in micropores is normalized to the total monolith volume (to compare with the density of interfibrillar water), adsorbed water contributes about 100 kg/m^3 below the *pseudo*-boiling temperature. At higher temperature gradual desorption of micropore water reduces its local density in micropores as the temperature increases. Supplementary Information presents all this information in detail.

Reviewer's comment: 4. Expression of "Density fluctuations induced by carbon porous material" is not appropriate. The density fluctuations should come from the phase behavior above the critical point. I wonder if the carbon porous materials just give the supercritical water which can be detected clearly by neutron imaging.

Authors' Response: We revised the text and removed this inappropriate expression. Indeed, our DA model does not indicate any possibility of water desorption from micropores at the high pressures used in the experiment, and before the *pseudo*-boiling temperature. Microporous carbons does not release adsorbed water before *pseudo*-boiling conditions. Moreover, it is shown in the revised manuscript that we recorded clear neutron images for the monolith-free case also, but they do not show the phase separation visible only in presence of the monolith.

Reviewer's comment: 5. Could you check the IUPAC recommendation on pore classification? Pure Appl Chem. 2015. Ultramicropores; pore width < 0.7 nm, Supermicropores; 2 nm > pore width >0.7 nm

Authors' Response: We thank the reviewer and have corrected the error.

Reviewer's comment: 6. Can you estimate density (GL) and density (LL) by use of the apparent density of activated carbon monolith and pore volumes of micropores and (mesopores) ? You can estimate the adsorption amount of water in micropore. If water in the micropores is predominant, you could estimate the densities of LL and GL water. This sounds important.

Authors' Response: We thank again the reviewer for this suggestion. Using the density and volume calculations we estimated that the fraction of water in micropores is not higher than 0.15 relative to the total water present in the monolith, which contains a large fraction of "free" water in the large interfibrillar space. To recall, the water present in micropores is not released until the *pseudo*-boiling temperature has been reached at each pressure. SI at section "Analysis of water density in activated carbon fiber micropores" presents the details.

Reviewer's comment: 7. T1, T2, and T3 are not clearly shown in Fig. 3

Authors' Response: The reviewer is right, and this comment is appreciated. We have simplified data presentation in the revised manuscript. The significant temperature in our revised manuscript is the *pseudo*-boiling temperature on the coexistence line above the supercritical point.

Reviewer's comment: 8. Hysteresis behaviors in Fig.4 are interesting. A more clear explanation is preferable.

Authors' Response: We observe this behavior only in the experiments with the monolith, therefore the fluid-solid interaction is contributing to this behavior. On the side of the solid monolith, we have a huge hydrophobic surface. On the side of the fluid, we have the thermodynamic transition from LL to GL, the *pseudo*-boiling, reflected macroscopically by the drop in density. Another physical phenomenon, related to the LL - GL changes is the dynamic transition, which is reflected by the changes in molecular motion between repulsion and attraction-dominated interaction. A hydrophobic surface is repulsive for LL water, which is polar, and clustering of water molecules via hydrogen bonding is likely to occur. The coalescence (attraction) of these clusters further increases the density, and this is the reason why in the after-monolith region we observe higher densities. In contrast,

GL water is non-polar, and carbon hydrophobic surface has affinity for water in GL state. The interplay between repulsion and attraction-dominated interactions might be responsible for the hysteresis behavior we observed, independently of pressure, and only in the experiments with the monolith.

REVIEWERS' COMMENTS:

Reviewer #1 (Remarks to the Author):

I appreciate that the authors addressed all the points raised in the previous reports. I think the authors did a great job and that the manuscript has improved considerably.

Including the density histogram in Fig. 2e is helpful in understanding the definitions for the densities of the GL and LL phases. I am also glad that the authors confirm that the pressure gradient through the monolith is very small and that the amount of water within the micropores (nanopores, indeed) is small relative to the water content in the inter-fiber voids. The addition of results in the absent of the monolith is very useful and strengthens the conclusions of this work.

I recommend the manuscript for publication in Nature Communications.

Reviewer #2 (Remarks to the Author):

Authors have revised the manuscript well on taking into account reviewer's comments. The manuscript including figures has been highly improved for better understanding. This paper is now worthy of publication, giving a great impact on supercritical water science.